# Genetic Variants Identified by Whole Exome Sequencing in a Large Italian Family with High Plasma Levels of Factor VIII and Von Willebrand Factor

**DOI:** 10.3390/ijms241814167

**Published:** 2023-09-15

**Authors:** Silvia Spena, Andrea Cairo, Francesca Gianniello, Emanuela Pappalardo, Mimosa Mortarino, Isabella Garagiola, Ida Martinelli, Flora Peyvandi

**Affiliations:** 1Fondazione IRCCS Ca’ Granda Ospedale Maggiore Policlinico, Angelo Bianchi Bonomi Hemophilia and Thrombosis Center, 20122 Milan, Italy; silvia.spena@policlinico.mi.it (S.S.); andrea.cairo@policlinico.mi.it (A.C.); francesca.gianniello@policlinico.mi.it (F.G.); mimosa.mortarino@policlinico.mi.it (M.M.); isabella_garagiola@yahoo.it (I.G.); ida.martinelli@moncucco.ch (I.M.); 2Department of Pathophysiology and Transplantation, Università degli Studi di Milano, 20122 Milan, Italy; emanuela.pappalardo@unimi.it

**Keywords:** high-throughput DNA sequencing, von Willebrand factor, factor VIII, microRNAs, thrombosis

## Abstract

High plasma levels of factor VIII (FVIII) and von Willebrand factor (VWF) have been indicated as independent risk factors for venous thromboembolism. However, the genetic factors responsible for their increase remain poorly known. In a large Italian family with high FVIII/VWF levels and thrombotic episodes, whole exome sequencing (WES) was performed on 12 family members to identify variants/genes involved in FVIII/VWF increase. Twenty variants spread over a 8300 Kb region on chromosome 5 were identified in 12 genes, including the low frequency rs13158382, located upstream of the *MIR143/145* genes, which might affect miR-143/145 transcription or processing. The expression of miR-143/145 and *VWF* mRNA were evaluated in the peripheral blood mononuclear cells of six family members. Members with the variant (n = 3) showed lower levels of both miRNAs and higher levels of *VWF* mRNA compared to members without the variant (n = 3). An analysis of genetic and expression data from a larger cohort of individuals from the 1000 Genomes and GEUVADIS project confirmed a statistically significant reduction (*p*-value = 0.023) in miR-143 in heterozygous (n = 35) compared to homozygous wild-type individuals (n = 386). This family-based study identified a new genetic variant potentially involved in VWF increase by affecting miR-143/145 expression.

## 1. Introduction

Factor VIII (FVIII) is a plasma glycoprotein predominantly synthesized in the liver by sinusoidal endothelial cells that circulates in a stable complex with von Willebrand factor (VWF), a large multimeric glycoprotein synthesized by endothelial cells and megakaryocytes. It is constitutively secreted and released upon stimulation by endothelial cells and secreted by activated platelets [1]. FVIII is involved in the intrinsic coagulation pathway by promoting the activation of factor X. VWF plays a dual role in hemostasis by supporting the adhesion and cohesion of platelets by acting as a ligand for their glycoproteins and preventing the degradation of plasma FVIII. The levels of FVIII and VWF in plasma are correlated, and both have been independently associated with a higher risk of venous thromboembolism (VTE) [2,3,4,5,6].

VTE is a multifactorial disease that develops as the result of three major mechanisms: blood stasis, endothelial damage, and hypercoagulability [7,8]. Several environmental risk factors are known to predispose VTE [9], while genetic risk factors predisposing VTE are partially known. Inherited risk factors that cause a hypercoagulable state, commonly known as thrombophilia abnormalities, include factor V Leiden (rs6025), the prothrombin variant rs1799963, and deficiencies of the natural anticoagulants antithrombin, protein C, and protein S. In addition, several genes mainly related to erythrocytes, platelets, and inflammation, but also including *F8* and *VWF*, that code for FVIII and VWF have been associated with the risk of VTE in meta-analyses of genome-wide association studies [10,11].

Plasma FVIII coagulant activity (FVIII:C) >150 UI/dL has been associated with an increased risk of VTE in a dose-dependent manner [12,13]. Moreover, high FVIII levels have been associated with an increased risk of VTE recurrence [2,14,15]. Thus, plasma levels of FVIII are sometimes measured as part of thrombophilia screening. High levels of FVIII may be due to congenital or acquired conditions (e.g., systemic inflammation). Only a few family-based studies have investigated the genetic factors responsible for FVIII increase [16], and the mechanism of this association is not well characterized. Recently, a prospective study confirmed the association between an increased risk of unprovoked VTE and high plasma levels of VWF [6]. In healthy individuals, plasma levels of VWF vary greatly, and approximately 65% of this variability is inherited [17] (with levels being regulated by several genetic loci) [18,19]. In this study, whole exome sequencing (WES) was performed on a large Italian family with high/extremely high levels of FVIII and VWF and episodes of thrombosis to identify new genetic risk factors associated with FVIII/VWF increase.

## 2. Results

### 2.1. Patient Data

In March 2016, an Italian woman (proband; II-8) was referred to the Angelo Bianchi Bonomi Hemophilia and Thrombosis Center (Milan, Italy) after a distal deep vein thrombosis of the left leg. Over the next 4 months, the proband suffered from two additional episodes of left-leg superficial thrombophlebitis. All the episodes were successfully treated with low-molecular-weight heparin. Thrombophilia screening performed two weeks after the second episode demonstrated very high levels of FVIII:C (>400%) (Figure 1, Table 1). In three subsequent measurements performed during the 7-month follow-up period, high or extremely high plasma levels of FVIII:C (up to 549%), FVIII antigen (FVIII:Ag) (520%), VWF activity (VWF:RCo) (209%), and VWF antigen (VWF:Ag) (308%) were also found (Table 1). FVIII and VWF were re-measured after 3 and 5 years. The levels decreased but still remained high (Table 1). The activity of the VWF-cleaving protease ADAMTS13 was slightly high (167%), while normal levels of anticoagulant protein S (105%), protein C (106%), and antithrombin (94%) were measured.

The family’s medical history revealed that three family members of the proband had had venous (II-3, II-5) and arterial (II-9) thrombosis (Table 1). Factor V Leiden (C>T) and prothrombin variant (G>A) genotyping performed on family members with available DNA (II-2, II-3, II-5, II-6, II-8, II-9, III-1, III-2, III-3, III-5, III-7, III-8) showed homozygous wild-type alleles (CC/GG) in all individuals except one (II-9) who was heterozygous for the prothrombin variant (CC/GA). Extremely high levels of FVIII:C (up to 397%), FVIII:Ag (up to 458%), VWF:RCo (up to 307%), and VWF:Ag (up to 390%) were also found in two proband’s siblings (II-3, II-5). Furthermore, high FVIII:C levels (up to 200%) and VWF:Ag (up to 227%) were measured in three family members (II-2, II-9, III-3), and four additional members (III-4, III-5, III-6, III-9) showed FVIII levels that were slightly above the upper limit of the normal range (Figure 1, Table 1).

In this family, additional disorders that may be associated with FVIII/VWF increase were reported, including the following: obesity (II-8, II-3, II-9), overweight (II-2, II-5, III-5), diabetes (II-8, II-2, II-3, II-5), dyslipidemia (II-8), atrial fibrillation (II-3), tumor and hypertension (II-5) (Table 1). Thus, the effect of these comorbidities, age, and blood group on high or extremely high FVIII and VWF levels was evaluated. For each variable, a positive relationship was found with extremely high levels of FVIII:C (range: 317–400%) and VWF:Ag (range: 295–390%) but not with high levels of FVIII:C (range: 180–200%) and VWF:Ag (range: 185–227%) (Appendix A). Thus, in the family, the presence of high levels of FVIII/VWF in (i) young subjects with O blood group and no/few comorbidities (Appendix A) and in (ii) two consecutive generations suggested a possible genetic predisposition to high levels of both FVIII and VWF and an autosomal modality of inheritance of both phenotypes. Hence, a whole exome sequencing study was initiated.

### 2.2. WES Reveals Variants Associated with High Levels of FVIII and VWF

WES was undertaken and carried out on subjects with DNA available at the time of the analysis: the proband (II-8) and 11 family members (II-2, II-3, II-5, II-6, II-9, III-1, III-2, III-3, III-5, III-7, III-8) (Figure 1, Table 1). After quality control, data analysis revealed a total of 208,103 variants in 26,457 genes. Variants were filtered according to the plasma levels of FVIII:C or VWF:Ag. Two case–control association analyses were then performed, assuming an autosomal dominant inheritance model in both cases (high levels being present in two consecutive family generations).

The FVIII-based analytical approach, conducted in seven cases (II-2, II-3, II-5, II-8, II-9, III-3, III-5) with FVIII:C > 150% and five controls (II-6, III-1, III-2, III-7, III-8) with FVIII:C < 150% (Figure 1, Table 1), allowed for the identification of three intronic variants in the *CYFIP2* gene and one synonymous variant in the *FNDC9* gene (Table 2, part A) located in the *CYFIP2* intron 23. All the variants found in the heterozygous state in the cases but absent in the controls were localized in a 56 Kb region on chromosome 5 and were common SNPs with a minor allele frequency (MAF) ranging between 8 and 46% (Table 2, part A).

The VWF-based approach, performed on six cases (II-2, II-3, II-5, II-8, II-9, III-3) with VWF:Ag > 180% and six controls (II-6, III-1, III-2, III-7, III-5, III-8) with VWF:Ag < 180% (Figure 1, Table 1), allowed for the identification of 13 variants (7 intronic, 4 synonymous, and 2 missense) in 9 genes (*ARHGEF37*, *TCOF1*, *NDST1*, *CCDC69*, *SLC36A3*, *SPARC*, *GALNT10*, *LARP1*, and *CYFIP2*), 2 variants in the 5′UTR of *THG1L* gene, and 1 variant located upstream of the *MIR143/145* genes (Table 2, part B). All these variants, which were found in the heterozygous state in the cases but not in the controls, were spread in a ~8300 Kb region of chromosome 5; most of them are common (MAF ranging 5.4–29.5%), four have a low frequency (MAF 1.2–4.1%), one is rare (MAF 0.1%), and one is not reported in dbSNP (Table 2, part B).

**Table 2 ijms-24-14167-t002:** Variants identified via WES through (**A**) the FVIII-based approach and (**B**) the VWF-based approach.

**(A)**															
									**PredictSNP2**						
**Position ^a^**	**Variant**	**dbSNP ID**	**MAF ^b^**	**Gene**	**RefSeq**	**cDNA**	**Protein**	**Type of Variant**	**CADD**	**DANN**	**FATHMM**	**FunSeq2**	**GWAVA**	**Final Output**	**ClinVar**
156714137	G>A	rs2288068	0.093	CYFIP2	NM_001037333	c.207 + 20G>A	/	intronic	neutral	deleterious	neutral	deleterious	deleterious	neutral	-
156727692	T>C	rs3815829	0.082	CYFIP2	NM_001037333	c.388 − 31T>C	/	intronic	neutral	neutral	neutral	-	neutral	neutral	-
156766037	A>G	rs3734027	0.465	CYFIP2	NM_001037333	c.2386 − 28A>G	/	intronic	deleterious	neutral	-	deleterious	neutral	deleterious	-
156770209	G>A	rs10214194	0.229	FNDC9	NM_001001343	c.336C>T	p.S112=	synonymous	deleterious	neutral	neutral	neutral	neutral	neutral	-
**(B)**															
									**PredictSNP2**						
**Position ^a^**	**Variant**	**dbSNP ID**	**MAF ^b^**	**Gene**	**RefSeq**	**cDNA**	**Protein**	**Type of Variant**	**CADD**	**DANN**	**FATHMM**	**FunSeq2**	**GWAVA**	**Final Output**	**ClinVar**
148808474	C>T	rs13158382	0.037	MIR143/145	NR_029684	c.−112C>T	/	upstream	deleterious	deleterious	neutral	deleterious	neutral	neutral	-
149003532	C>A	-	-	ARHGEF37	NM_001001669	c.1336 − 43C>A	/	intronic	deleterious	neutral	-	-	deleterious	neutral	-
149776232	C>T	rs15251	0.219	TCOF1	NM_001135243	c.4169C>T	p.A1390V	missense	neutral	deleterious	neutral	neutral	-	neutral	benign
149776355	G>C	rs45491898	0.016	TCOF1	NM_001135243	c.4292G>C	p.G1431A	missense	neutral	deleterious	neutral	neutral	-	neutral	likely benign
149919778	G>A	rs61732050	0.041	NDST1	NM_001543	c.1701G>A	p.A567=	synonymous	neutral	neutral	neutral	neutral	neutral	neutral	-
149932712	G>A	rs62383060	0.001	NDST1	NM_001543	c.2530 − 63G>A	/	intronic	deleterious	neutral	neutral	neutral	neutral	neutral	-
150578574	A>G	rs3734038	0.195	CCDC69	NM_015621	c.303T>C	p.N101=	synonymous	neutral	neutral	neutral	neutral	neutral	neutral	-
150603444	C>G	rs248461	0.295	CCDC69	NM_015621	c.−40487 + 40G>C	/	intronic	neutral	neutral	neutral	deleterious	neutral	neutral	-
150666933	C>A	rs375396	0.218	SLC36A3	NM_001145017	c.705G>T	p.L235=	synonymous	neutral	neutral	neutral	neutral	neutral	neutral	-
151046928	C>T	rs729853	0.168	SPARC	NM_003118	c.585 + 100G>A	/	intronic	neutral	neutral	neutral	-	neutral	neutral	-
151054137	A>C	rs2116780	0.170	SPARC	NM_003118	c.120 + 36T>G	/	intronic	neutral	neutral	neutral	-	neutral	neutral	-
153783753	C>T	rs6580076	0.242	GALNT10	NM_198321	c.1146C>T	p.A382=	synonymous	neutral	neutral	neutral	neutral	neutral	neutral	-
154135386	G>A	rs78112077	0.054	LARP1	NM_015315	c.206 − 34499G>A	/	intronic	neutral	deleterious	-	neutral	neutral	deleterious	-
156816521	C>T	rs142867180	0.012	CYFIP2	NM_001037333	c.3446 + 86C>T	/	intronic	deleterious	neutral	neutral	deleterious	neutral	neutral	-
157158414	C>A	rs2270819	0.115	THG1L	NM_017872	c.−35C>A	/	5′UTR	neutral	deleterious	deleterious	-	deleterious	deleterious	-
157158439	C>A	rs2270818	0.115	THG1L	NM_017872	c.−10C>A	/	5′UTR	deleterious	neutral	deleterious	deleterious	deleterious	deleterious	-

Abbreviations: MAF, minor allele frequency; CADD, Combined Annotation Dependent Depletion; DANN, Deleterious Annotation of Genetic Variants using Neural Networks; FATHMM, Functional Analysis Through Hidden Markov Models; GWAVA, Genome-Wide Annotation of Variants; HGMD, Human Gene Mutation Database. The “-” denotes a variant not properly evaluated via the means of a specific tool. ^a^ Genomic coordinates of chromosome 5 (hg19). ^b^ Frequency from 1000 Genomes Project Phase 3_European population.

### 2.3. In Silico Prediction of Variants Identified by WES

The PredictSNP2 consensus classifier, which combined the five best performing prediction methods (CADD, DANN, FATHMM, FunSeq2, and GWAVA) to provide a prediction on the pathogenicity of the variants, was assessed. Among the four variants identified via the FVIII:C approach, only the rs3734027 variant was predicted to be deleterious (Table 2, part A). This is an A to G transition localized at -28 nucleotides from the acceptor splice site (3′ss) of the *CYFIP2* exon 21. This A>G nucleotide substitution was predicted by the Human Splicing Finder tool (http://www.umd.be/HSF3/, accessed on 1 March 2019) to reduce the branch point score from 95.75 to 66.12 (Appendix A). Among the variants identified by the VWF:Ag approach, the deep intronic variant rs78112077 in *LARP1* and both rs2270818 and rs2270819 localized in the 5′UTR of *THG1L* were predicted to be deleterious (Table 2, part B). The rs78112077 variant is a G to A transition that is localized in the exon 1 of the *LARP1* alternative transcripts NM_001367718 and NM_033551 (Appendix A), resulting in the synonymous Gly23 = substitution (Appendix A), was also predicted via the use of the NNSPLICE tool (https://www.fruitfly.org/seq_tools/splice.html, accessed on 15 February 2022) to activate an intronic cryptic donor splice site (5′ss) (score 0.75) localized 369 nt upstream of the physiologic exon 1 5′ss (score 0.93) of the *LARP1* alternative transcripts NM_001367718 and NM_033551 (Appendix A).

No deleterious effect was predicted by PredictSNP2 for the two rare intronic variants in the *ARHGEF37* and *NDST1* genes (Table 2, part B).

The two missense variants (p.A1390V, rs15251 and p.G1431A, rs45491898) in *TCOF1* were classified as benign and likely benign by ClinVar (https://www.ncbi.nlm.nih.gov/clinvar/, accessed on 28 January 2019) (Table 2, part B).

The VarElect tool was also used to prioritize genes identified via WES analysis. A total of nine genes with a direct (seven) or indirect (two) link with thrombosis, inflammation, and VWF were found (Table 3). Among them, the *MIR143*, *MIR145,* and *SPARC* genes had the higher score (range: 5.38–8.45) (Table 3). No pathogenic role was predicted for variants rs729853 and rs2116780 in the *SPARC* gene (Table 2), and no correlation with the chosen phenotypes was found for the *CTB* gene, the putative target of the regulatory elements (promoter and enhancer) where these variants are localized (Table 3). Conversely, the rs13158382 variant, mapped via EnhancerAtlas in the common promoter of *MIR143/145* genes (Table 2, part B), was localized at −122 nucleotides from the mature miR-143 and −7 nucleotides from the precursor of miR-143 with a possible role on miRNAs processing/expression.

Taken together, these findings prompted us to further investigate the predicted pathogenic role of the two intronic variants (rs3734027 and rs78112077) in *CYFIP2* and *LARP1*, the two variants (rs2270818 and rs2270819) in the 5′UTR of *THG1L*, and the rs13158382 variant located upstream the *MIR143/145* in vitro.

### 2.4. Analysis of Intronic Variants in CYFIP2 and LARP1

The effect of the *CYFIP2* rs3734027 (A>G) and *LARP1* rs78112077 (G>A) was evaluated via reverse transcription (RT)-polymerase chain reaction (PCR) on total RNA from peripheral blood mononuclear cells (PBMCs). An analysis of the *CYFIP2* transcript in the heterozygous (AG) proband and two homozygous (AA and GG) subjects showed two amplicons of different intensity for all three analyzed samples (Appendix A). Direct sequencing of the strong high band (399 bp) showed expected splicing, while the low-light band (199 bp) resulted from the skipping of exon 21. Quantitative polymerase chain reaction (qPCR) analysis showed no increase in the skipped isoform in individuals with the G allele (Appendix A). Our analysis of *LARP1* alternative transcripts in the heterozygous (GA) proband and a homozygous (GG) subject showed no amplification of *LARP1* in both samples despite the amplification of the housekeeping *GAPDH* transcript, thus suggesting a possible lack of expression of the alternative transcripts in the analyzed cells.

### 2.5. Analysis of 5′UTR Variants in THG1L

Expression of *THG1L* was evaluated in a larger cohort of subjects with and without the rs2270818 (C>A) and rs2270819 (C>A) variants by retrieving RNA sequencing data available at the GEUVADIS consortium. No statistically significant difference in gene expression was found among the groups of compound heterozygotes (CA/CA) and homozygotes (CC/CC; AA/AA).

### 2.6. Analysis of the Variant in the Promoter of MIR143/145

The effect of the rs13158382 (C>T) variant on levels of miR-143 and miR-145 was first evaluated in three heterozygous (CT) (II-8, II-5, II-9) and three homozygous (CC) (III-4, III-6, III-9) family members. In them, levels of *VWF* and *F8* mRNAs were also analyzed. A slight reduction in both miR-143 and miR-145 and a slight increase in *VWF* compared to homozygotes was observed in the heterozygous subjects (Figure 2A–C, respectively), while no difference regarding *F8* was observed between the two groups (Figure 2D). The observed differences did not reach statistical significance, perhaps due to the very small number of analyzed individuals. To confirm the putative effect of rs3158382 on miR-143 and miR-145 expression in a larger cohort, the genetic and miRNA profiling data of 452 subjects from the GEUVADIS project were obtained and analyzed. A total of 31 subjects—27 homozygous (CC) and 4 heterozygous (CT)—who exceeded the first or the third interquartile were excluded from the analysis as outliers. Due to the non-normal distribution of miRNA expression values (Shapiro–Wilk: *p*-value = 0.03653), the nonparametric test (Mann–Whitney U) was performed and confirmed the statistical significance (*p*-value = 0.023) of the reduction in miR-143 expression among the heterozygous subjects compared to the homozygous wild-type ones (Figure 3A). The expression levels of miR-145 were too low to allow for a differential expression analysis. The mRNA data of *VWF* and *F8* were also retrieved and analyzed using the same statistical approach; levels of *VWF* expression were too low (mean count of unnormalized matrix < 7) to allow a differential expression analysis, and no difference in the expression of *F8* was found (Figure 3B).

### 2.7. Impact of the rs13158382 Variant on miRNA Synthesis and Processing

Additional investigations were performed in order to assess whether the reduction in mature miRNAs resulted from a lowered synthesis of primary miRNA (pri-miRNA) and/or from the aberrant processing of precursor miRNA (pre-miRNA). Levels of pri-miRNA-143/145, evaluated in heterozygous (CT) (II-5, II-9) and homozygous (CC) (III-4, III-6, III-9) family members were slightly reduced in heterozygotes compared to homozygotes (Appendix A), thus suggesting that the rs13158382 variant had an impact on miRNA synthesis. The impact of the rs13158382 variant on the secondary structure of the pre-miR-143 was also evaluated using the miRVaS tool. Three different representations of secondary structures (centroid, maximal expected accuracy, and minimal free energy) were accessed. All strategies predicted structural changes in the hairpin and flanking regions that may affect miRNA processing, such as an alteration of Drosha cleavage site (Figure 4). The effect of rs13158382 on the secondary structure of miR-145 could not be predicted since this variant is located 1735 nucleotides upstream of the pre-miRNA sequence.

## 3. Discussion

A large Italian family presenting with high/extremely high levels of both FVIII and VWF was investigated to identify new genetic factors predisposing to FVIII/VWF increase. Since, in the general population, VWF:Ag and FVIII:C levels are regulated by genetic (blood group) and acquired factors (comorbidities, body mass index, age) [20], the putative contribution of these variables on the levels of FVIII and VWF was evaluated in the family. The expected correlation was found in subjects with normal and extremely high levels of FVIII/VWF (>500% the lower limit of normal ranges). Conversely, no correlation was found in family members with high levels (up to 300% the lower limit of normal ranges) (Appendix A). These observations suggested the presence of a possible genetic predisposition to high FVIII/VWF levels, coupled with an additive effect of non-O blood group, elderly age, and other acquired variables on the additional FVIII/VWF increase in family members with extremely high levels of VWF. Levels (normal/high/extremely high) of FVIII and VWF were positively correlated in the majority (11) of family members; only 4 members (III-4, III-5, III-6, III-9) had FVIII levels slightly above the upper normal limit and VWF levels in the normal range. Since member III-3 had FVIII levels similar to those of member III-5 (200% vs. 198%) but higher levels of VWF (227% vs. 145%) (Figure 1, Table 1), the reduced levels of VWF found in III-5 could be assay-related or partially due to the different blood groups (non-O vs. O) that are known to contribute to ~15% of the VWF:Ag variance in plasma [21]. In our family, high levels of FVIII and VWF were found in two consecutive generations, suggesting an autosomal dominant inheritance of both phenotypes (Figure 1). Hence, two (FVIII and VWF based) case–control association analyses were performed on WES data, leading to the identification of two overlapping regions on chromosome 5 (56 Kb at q32 and 8350 Kb at q32-q33.3, respectively) (Table 2). This result is not surprising because FVIII is bound by VWF, and the plasma levels of the two proteins are closely related. Indeed, VWF deficiency is accompanied by low levels of FVIII, and, in turn, increased VWF levels induce an increase in FVIII [22,23]. In addition, previous genetic association studies identified and replicated few loci associated with FVIII plasma levels that were a subset of 15 loci spread on different chromosomes associated with VWF plasma levels [18,24,25,26]. Concerning chromosome 5, two genes (*TMEM171* and *TNPO1*) localized at q13.2 have been functionally characterized in vitro, and their silencing was found to increase the release of VWF by HUVEC cells [18].

Among the genes identified by our analysis, only *NDST1*, *CYFIP2*, and *MIR143/145* have a putative functional link to VWF. *NDST1* is involved in the differentiation of embryonic stem cells into endothelial cells, with transcript and protein levels of *NDST1* being directly correlated to the levels of VWF in endothelial cells [27]. CYFIP2 protein is a component of the WAVE1 complex; the binding of WAVE1 to RAC1 mediates actin polymerization, and the depletion of RAC1 has been demonstrated to prevent VWF secretion in HUVEC cells [28]. Concerning miR-143 and miR-145, their downregulation has been associated with the upregulation of the VWF protein in patients with acute coronary syndrome and of the *VWF* transcript during the endothelial differentiation of human adipose-derived stem cells (hADSCs) [29,30,31]. It is worth noting that individuals in the family under study with reduced levels of miR-143/145 and high (II-9)/extremely high (II-3, II-8) levels of VWF suffer from a different type of obesity (Table 1). miR-143 and miR-145 have also been linked to thrombosis. In particular, the intravenous injection of miR-145-transfected endothelial progenitor cells (EPCs) facilitates the recanalization of arterial thrombi [32]. Coagulation factor XI and tissue factor (TF) are targets of miR-145 [33,34]; in patients with VTE, both proteins were elevated, and an inverse correlation between miR-145 levels and TF levels was observed [34]. Moreover, the restoration of miR-145 levels in thrombotic rats via miR-145 mimic delivery resulted in decreased TF level and activity, accompanied by reduced thrombogenesis [34]. Since TF is a potent stimulator of VWF secretion [35], a possible link between the reduction in miR-143/145 and increase in VWF mediated by TF may be hypothesized, but TF measurements were not performed for our family members.

Low levels of miR-143 and miR-145 are also associated with metabolic diseases such as type 2 diabetes, atherosclerosis, obesity, and an increased inflammatory response [36,37,38]. Recently, it has been reported that miR-145, which is secreted by mesenchymal-stem-cells, packaged in exosomes, and delivered to endothelial cells, reduces the formation of atherosclerotic plaques in mice [39]. It is worth noting that several diseases, such as diabetes, obesity, cancer, and hypertension, were present in our family members.

Very recently, experiments in HEK293 and HUVEC cell lines also demonstrated that miR-143 could target and inhibit *VWF* [40].

In vitro and in silico studies on the identified variants highlighted a possible causal role only for the low-frequency rs13158382 variant (located upstream of the *MIR143/145* genes). The *MIR143/145* genes clustered at 5q32 share a common promoter consisting of a conserved region of 4.2 Kb that regulates their expression. miR-143 and miR-145 are co-transcribed into a single bicistronic unit consisting of a primary transcript (pri-miRNA) [41] with a hairpin structure and are released into the nucleus as precursor miRNA (pre-miRNA) after Drosha and DGCR8 cleavage [42]. The identified rs13158382 variant is a low-frequency SNPs (MAF 0.037) located 7-bp from pre-miRNA that has been previously predicted to modify a transcription factor binding site [43]. Two other SNPs (rs4705342 and rs4705343), located in the promoter of *MIR143/145* at −400 and −510 bp, respectively, have been functionally characterized via dual-luciferase reporter assays and have been found to increase and reduce luciferase activity respectively, thus suggesting altered promoter activity and miR-143/145 synthesis [44,45], as observed in our family (Appendix A).

Since genetic variants in or close to miRNA genes can affect structural modifications [46], we also assessed the impact of the rs13158382 variant on the secondary structure of pre-miR-143 using the mirVAS tool. A conformational change in the miRNA secondary structure was predicted, with potential consequences on mature miRNA processing.

To assess the (inverse) correlation between miR-143/145 and *VWF* transcript levels, we performed transcript analysis in the available cells (PBMCs) of family members with and without the rs13158382 variant. The expected trend was observed, even if differences were not statistically significant, perhaps due to the small number of analyzed samples and low (i.e., ectopic) expression of *F8/VWF* in PBMCs. To further tackle this issue, we analyzed miR-143, miR-145, *F8*, and *VWF* transcript data from a larger cohort of subjects [47]. A differential expression analysis, feasible only for miR-143 and *F8*, confirmed a statistically significant reduction in miR-143 in heterozygous rs13158382 individuals, thus suggesting a possible role for the identified variant in miR-143 expression. We assessed BioGPS, Geo Profiles, and ImmGen Project databases containing expression data [48], but no genetic and transcriptional evidence concerning cells expressing the *VWF* were available for further analyses.

The limitations of this study are attributable to the following: (i) the unavailability of endothelial cells to evaluate the real expression of *F8*/*VWF* transcripts and (ii) the lack of in vitro studies to confirm the role of the rs13158382 variant on miR-143/145 transcription or processing. Hence, a luciferase assay should be performed to confirm our findings. In conclusion, notwithstanding the aforementioned limitations, there is evidence that the rs13158382 variant modulated miR-143/145 expression, even though deeper investigations are needed in order to understand the effective contribution of this genetic variant to miRNA expression and to elucidate new possible molecular mechanisms underlying *VWF* expression and VTE. If confirmed, the rs13158382 variant may become a new genetic risk factor of VTE and miR-143/145 potential biomarkers of VTE and a potential future therapeutic tool to reduce *VWF* expression.

## 4. Materials and Methods

### 4.1. Patients and Blood Samples

A 51-year-old Italian woman (proband; II-8) and a total of 14 proband’s family members (5 siblings: II-2, II-3, II-5, II-6, II-9; 2 children: III-7 III-8; and 7 nieces/nephews; III-1, III-2, III-3, III-4, III-5, III-6, III-7, III-8, III-9) were investigated (Figure 1). All individuals signed an informed consent form prior to blood sampling. A total of 10 mL of whole blood were collected in sodium citrate tubes for plasma separation and DNA extraction. In addition, 2.5 mL of whole blood were collected in PaxGene^®^ tubes (BD, Franklin Lakes, NJ, USA) for RNA extraction.

### 4.2. Plasma Measurements

Plasma was obtained via the centrifugation of sodium citrate tubes at 3000 rpm for 15 min at room temperature. FVIII:C and FVIII:Ag were measured via a one-stage clotting assay [49] and via enzyme immunoassay, respectively, using the Asserachrom VIII:Ag kit (Stago, Asnières sur Seine, France) (normal range for both tests: 51–147%). VWF:RCo and VWF:Ag were evaluated using the von Willebrand Ristocetin Cofactor (normal range 41–160% and 53–168% for O and non-O blood groups, respectively) and an automated immunoassay on ACL TOP analyzer (Instrumentation Laboratory, Milan, Italy), respectively (normal range: 40–165% and 55–169% for O and non-O blood groups, respectively). ADAMTS13 activity was measured using FRETS-VWF73 [50] (normal range: 45–138%). The activity of antithrombin (AT), protein C (PC), and protein S (PS) were measured via chromogenic (AT, PC) and clotting (PS) assays (Instrumentation Laboratory, Milan, Italy) using an ACL TOP analyzer (normal range: 82–112%, 65–160%, and 58–114%, respectively).

### 4.3. DNA and RNA Extraction

Genomic DNA was extracted from PBMCs via the standard salting-out method [51].

Total RNA, including miRNA, was extracted from whole blood using the PaxGene^®^ Blood miRNA kit (Preanalytix, Zurich, Switzerland), following manufacturer’s protocol. The DNA and RNA samples were quantified using the NanoDrop 2000c spectrophotometer (Thermo Fisher Scientific, Wilmington, DE, USA). A total of 15 DNA samples (12 collected before and 3 after the whole exome sequencing) and 6 RNA samples were available for study.

### 4.4. WES and Data Analysis

WES was performed on proband and 11 family members (II-2, II-3, II-5, II-6, II-9, III-1, III-2 III-3, III-5, III-7, III-8) (Figure 1). Library preparation was performed using the SureSelect Human All Exon kit (Agilent Technologies, Santa Clara, CA, USA) according to the manufacturer’s instructions. The sequencing of paired-end fragments was conducted on an Illumina HiSeq 2500 sequencing platform (Illumina, San Diego, CA, USA) at the Genewiz (Genewiz, South Plainfield, NJ, USA).

Raw sequencing data were analyzed according to the guidelines of the Broad Institute (https://software.broadinstitute.org/gatk/best-practices/, accessed on 25 January 2019) [52]. The quality of the FASTQ files was verified using FastQC v0.11.5 (Babraham Institute, Cambridge, UK) [53]. The reads were aligned to the reference genome (GRCh37/hg19) using the Burrows–Wheeler Aligner v0.7.17 [54], and indel realignment was conducted using the GATK toolkit v4.0.12.0. The bed file for the analysis was padded with 100 nucleotides. Variants were called using the haplotype caller (GATK), and the obtained VCF file was annotated via the use of Annovar (https://annovar.openbioinformatics.org/en/latest/, accessed on 28 January 2019) [55] and KGGseq v.1.0 [56]. Variants with allele numbers > 2, a lack of the vcf filter in PASS, and a sequencing quality score < 30 were removed. Genotypes with depth reads < 10 and a quality score < 20 were excluded from analysis.

High levels of FVIII and VWF were considered as two different phenotypes, and variant filtering was performed twice. Variants were filtered according to an autosomal dominant inheritance model of high levels of FVIII or VWF; variants observed to be in the heterozygous state in all family members with FVIII:C > 150% or VWF:Ag > 180% (i.e., cases) and absent in individuals with FVIII:C < 150% or VWF:Ag < 180% (i.e., controls) were further analyzed (FVIII and VWF cut-off values were chosen based upon literature data and the upper normal limits of FVIII/VWF measurement assays). Only variants with a call rate of 100% were included.

### 4.5. Prioritization of Variants and Genes

The effect of the identified variants was predicted in silico using the PredictSNP2 tool v2.1 [57]. The localization of variants in predicted regulatory regions was accomplished using the Enhancer Atlas (http://enhanceratlas.org/, accessed on 17 September 2021) database [58]. Genes were prioritized based on the phenotype of interest (i.e., thrombosis, VWF disease, and inflammation) using the VarElect tool (https://varelect.genecards.org/, accessed on 17 September 2021) [59]. Genotyping of cases and controls was performed via the direct sequencing of the regions encompassing the identified variants using the BigDye Terminator Cycle Sequencing Ready Reaction Kit (Applied Biosystems, Waltham, MA, USA) and an ABI PRISM 3130 Genetic Analyzer (Applied Biosystems). Primers and PCR conditions are available upon request.

### 4.6. First-Strand cDNA Synthesis and qPCR

First-strand cDNA synthesis was performed on total RNA by using random nonamers and the High-Capacity cDNA Reverse Transcription kit (Thermo Fisher, Waltham, MA, USA). miRNA-specific RT was performed on total RNA by using a specific RT primer and the TaqMan™ MicroRNA Reverse Transcription kit (Thermo Fisher).

The genotyping of factor V Leiden and the prothrombin variant was performed on 175 ng of DNA by using the Q-PCR Alert Kit (ELITechGroup, Berkhamsted, UK).

Expression analysis of *F8*, *VWF*, pri-miRNA-143/145, and the *TBP* housekeeping gene was performed on 50 ng of cDNA by using TaqMan™ Gene Expression Assays Hs00252034_m1, Hs01109446_m1, Hs03303166_pri/Hs03303169_pri, Hs00427620_m1, respectively (Thermo Fisher), and TaqMan Fast Advanced Master mix (Thermo Fisher). Expression analysis of miR-143 (hsa-miR-143-3p), miR-145 (hsa-miR-145-5p), and of the housekeeping RNU6B was performed on 50 ng of miRNA specific cDNA by using TaqMan™ MicroRNA Assays 002249, 002278, 001093, respectively (Thermo Fisher), and TaqMan Fast Advanced Master mix (Thermo Fisher).

The fluorescence signals were monitored using the StepOnePlus Real-Time PCR system (Applied Biosystems) and StepOne software v2.3 (Applied Biosystems). Two duplicates of each sample were analyzed, and the relative amounts were determined using the 2−∆∆Ct method. Student’s *t*-test was applied for the statistical analysis.

### 4.7. In Silico Analysis of Public Expression Data

RNAseq data available at the GEUVADIS consortium (https://www.internationalgenome.org/data-portal/data-collection/geuvadis, accessed on 20 September 2021) [47] and obtained from lymphoblastoid cell lines of 465 subjects enrolled in the 1000 Genomes project were analyzed. The relative normalized expression matrix of mRNA and miRNA were downloaded from the EBI Array Express archive (https://www.ebi.ac.uk/arrayexpress/files/E-GEUV-1/analysis_results/GD462.GeneQuantRPKM.50FN.samplename.resk10.txt.gz (accessed on 20 September 2021) and http://www.ebi.ac.uk/arrayexpress/files/E-GEUV-3/analysis_results/GD452.MirnaQuantCount.1.2N.50FN.samplename.resk10.txt, respectively, accessed on 20 September 2021). For each variant, the genotype of each subject was downloaded (https://www.ebi.ac.uk/arrayexpress/files/E-GEUV-1/genotypes/, accessed on 20 September 2021), and data were extracted using VCFtools v.0.1.16 [60]. Statistical analysis was performed using R-4.1.0. Outliers exceeding 1.5× the first or the third interquartile were excluded from the analysis. The Shapiro–Wilk test was used to check the expression value distribution. A non-parametric Mann–Whitney U-test was used to evaluate the differential expression of miRNAs. A *p*-value < 0.05 was considered statistically significant.

### 4.8. Secondary Structure Prediction

The impact of the genetic variant on miRNA secondary structure was predicted using the mirVAS tool (http://mirvas.bioinf.be/, accessed on 8 October 2021); the default setting (i.e., 100 nucleotides upstream and downstream of the pre-miR-143) was used for the analysis.

## Figures and Tables

**Figure 1 ijms-24-14167-f001:**
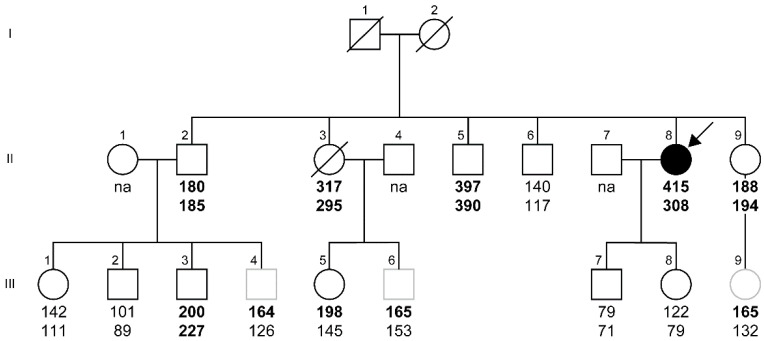
Pedigree of the family. Levels of FVIII coagulant activity (FVIII:C) and VWF antigen (VWF:Ag) are indicated below each symbol and reported as percentages (%) (FVIII:C normal range 51–147%; VWF:Ag normal range 40/55–165/169%). Values above the upper normal limits are indicated in bold. The arrow indicates the proband. Barred boxes refer to deceased family members, and gray boxes denote family members not tested by WES. na, not available.

**Figure 2 ijms-24-14167-f002:**
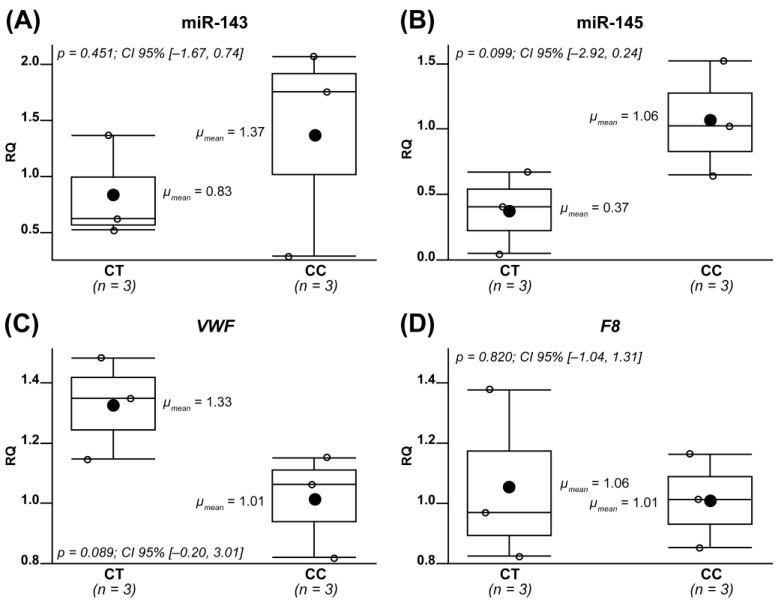
rs13158382 (C>T) variant and transcript levels in heterozygous and homozygous family members. Box plots show the relative quantitation (RQ) of (**A**) miR-143, (**B**) miR-145, (**C**), *VWF*, and (**D**) *F8* transcripts in heterozygous (CT) and homozygous (CC) family members. “n” denotes the number of analyzed individuals for each genotype.

**Figure 3 ijms-24-14167-f003:**
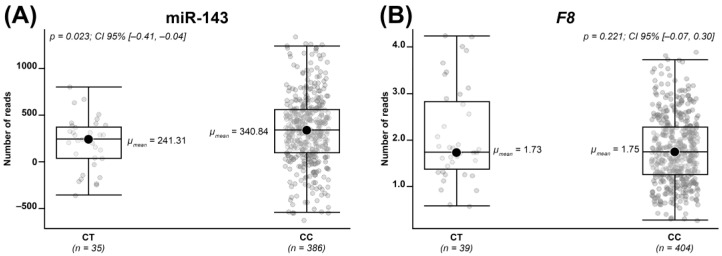
rs13158382 (C>T) variant and transcript levels in heterozygous and homozygous subjects from the GEUVADIS project. Box plots show the number of reads of (**A**) miR-143 and (**B**) *F8* evaluated in heterozygous (CT) and homozygous (CC) subjects from the GEUVADIS project. “n” denotes the number of analyzed individuals for each genotype.

**Figure 4 ijms-24-14167-f004:**
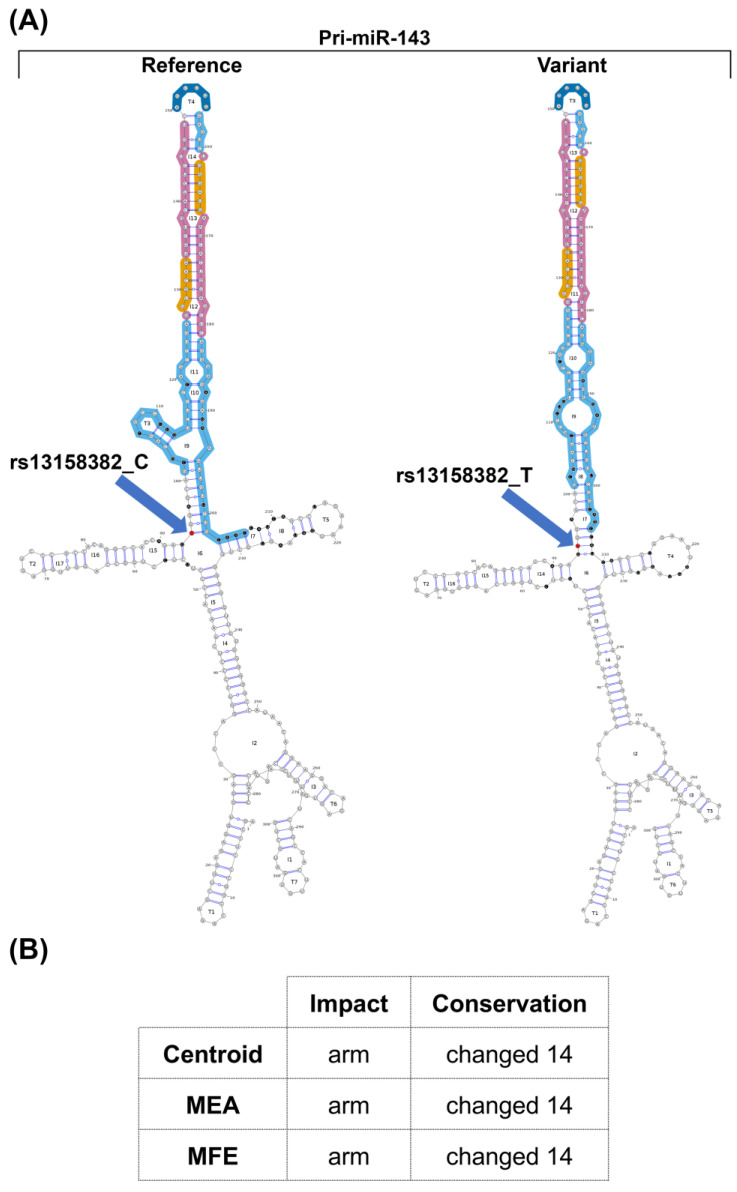
Prediction of miR-143 secondary structures. (**A**) miRVaS visual output shows the predicted structure of pri-miR-143 with (Variant) and without (Reference) the rs13158382 variant. A structural change in the 5p flanking region upstream of the hairpin arm is shown. Functional regions within the pre-miRNA are differentially colored: seed sequence (orange), mature miRNAs (magenta), terminal loop (blue), hairpin arm (cyan). The localization of the rs13158382 variant (red) and the nucleotides involved in putative structural changes (black) are highlighted. (**B**) miRVaS output shows the predicted most important region (arm) with a structural impact and the predicted conservation of the hairpin along with the number of changed bases within the hairpin. All predictions are based on minimal free energy (MFE), maximal expected base-pair accuracy (MEA), and the smallest total base-pair distance to the sampled structures of that ensemble (centroid).

**Table 1 ijms-24-14167-t001:** Clinical and laboratory data of analyzed subjects.

Subject ^a^	Sex	Age at First Visit	Blood Group	BMI	Thrombosis	Comorbidity ^b^	Therapy	FVIII:C (51–147%)	FVIII:Ag (51–147%)	VWF:Ag (40–165%) ^c^ (55–169%) ^d^	VWF:RCo (41–160%) ^c^ (53–168%) ^d^
Proband II-8	F	51	B	40.1	Deep vein thrombosis Superficial thrombophlebitis	Type III obesity Diabetes Dyslipidemia	Semaglutide Ezetimibe Vitamin D	415 389 398 549 239 269	- - - 520 - -	- - - 308 151 210	- - - 209 133 157
II-2	M	55	B	29.1	no	Overweight Diabetes	no	180	195	185	120
II-3	F	65	B	35.2	Pulmonary embolism Superficial thrombophlebitis Iliac artery stent	Type II obesity Diabetes Atrial fibrillation	Insulin Vitamin K antagonist	317	336	295	234
II-5	M	63	B	29.4	Superficial thrombophlebitis	Overweight Diabetes Hypertension Prostate carcinoma	Metformina Antihypertensive drug	397 225	458 -	390 210	307 157
II-6	M	57	O	19.0	no	HIV	Anti-retroviral drug	140	151	117	117
II-9	F	66	O	32.0	Iliac artery thrombosis	Type I obesity	Aspirin	188 189	194 -	194 203	145 176
III-1	F	12	B	-	-	-	-	142	-	111	103
III-2	M	19	B	-	no	no	no	101	-	89	72
III-3	M	10	B	-	-	-	-	200	-	227	146
III-5	F	31	O	25.7	no	Overweight	no	198	251	145	108
III-7	M	15	O	18.3	no	no	no	79	97	71	61
III-8	F	30	O	21.0	no	no	no	122	124	79	79
*III-4*	M	-	-	-	-	-	-	164	-	126	123
*III-6*	M	-	-	-	no	no	no	165	-	153	134
*III-9*	F	-	-	-	no	no	no	165	-	132	122

Abbreviations: F, female; M, male; BMI, body mass index; FVIII:C, factor VIII coagulant activity; FVIII:Ag, factor VIII antigen; VWF:Ag, von Willebrand factor antigen; VWF:RCo, von Willebrand Ristocetin cofactor. ^a^ Subjects are named according to the pedigree depicted in Figure 1; subjects not analyzed via WES are in italics. ^b^ Diseases that may be associated with an increase in FVIII/VWF levels. ^c^ Normal range for O blood type. ^d^ Normal range for non-O blood type. The “-” denotes unknown data.

**Table 3 ijms-24-14167-t003:** VarElect gene prioritization.

**Gene** **(Direct)**	**Category**	**Matched Phenotypes**	**Score** **(Direct Gene)**		
MIR145	RNA Gene	thrombosis, inflammation, VWF	8.45		
SPARC	Protein Coding	thrombosis, inflammation, VWF	5.60		
MIR143	RNA Gene	thrombosis, inflammation, VWF	5.38		
NDST1	Protein Coding	thrombosis, inflammation	0.80		
CYFIP2	Protein Coding	inflammation, VWF	0.64		
ARHGEF37	Protein Coding	VWF	0.15		
CCDC69	Protein Coding	inflammation	0.14		
**Gene** **(Indirect)**	**Category**	**Matched Phenotypes**	**Score** **(Indirect Gene)**	**Gene** **(Target)**	**Score** **(Target Gene)**
THG1L	Protein Coding	thrombosis, inflammation, VWF	2.51	IL10	1.81
THG1L	Protein Coding	thrombosis, inflammation, VWF	2.51	TP53	1.02
THG1L	Protein Coding	thrombosis, inflammation, VWF	2.51	CALR	0.49
THG1L	Protein Coding	thrombosis, inflammation, VWF	2.51	PTEN	0.33
THG1L	Protein Coding	thrombosis, inflammation, VWF	2.51	SERPINE1	0.27
CARMN ^a^	Protein Coding	thrombosis, inflammation, VWF	0.05	CSNK1A1	0.03
CARMN ^a^	Protein Coding	inflammation	0.05	IL17B	0.02
CARMN ^a^	Protein Coding	VWF	0.05	PCYOX1L	0.01
CARMN ^a^	Protein Coding	inflammation	0.05	GRPEL2	0.00

^a^ *MIR143* and *MIR145* genes are localized in an exonic region and an intronic region of the *CARMN* gene, respectively.

## Data Availability

Data sharing is not applicable.

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
