# Peer review of "Genetic Variants Identified by Whole Exome Sequencing in a Large Italian Family with High Plasma Levels of Factor VIII and Von Willebrand Factor"

_ijms, 2023, doi:10.3390/ijms241814167_

Round 1

Reviewer 1 Report

Authors conducted whole-exome sequencing to identify sequence variants based on the dichotomy of FVIII or VWF levels and measured RNA expression levels of the genes adjacent to the sequence variants in the peripheral blood mononuclear cells. They concluded that rs13158382 genetic variant is potentially involved in VWF increase by affecting miR-143/145 expression and may explain a large Italian family with high FVIII/VWF levels and thrombotic episodes. Although the findings are interesting, the role of the sequence variant to high FVIII or VWF levels, or thrombotic episodes has not yet been demonstrated. The RNA expression data in GEUVADIS were not incorporated with FVIII, VWF, or thrombotic episode information for the analyses. The relevance of VWF and/or miR-143/145 expressions in the peripheral blood mononuclear cells to those in the endothelial cells and megakaryocytes, that are the primary source of FVIII and/or VWF, has not been proven. In other words, VWF or miR-143/145 expression in the peripheral blood mononuclear cells has not yet been correlated with thrombotic episode. Measurements of VWF and miR-143/145 expressions in the endothelial cells and/or megakaryocytes are essential.

Many grammatical, typographic, and English expression errors were found.

Reviewer 2 Report

Excellent analysis with very interesting findings.  Just a few minor issues:

I would appreciate an explanation or theory of why miRN143/145 variants affect vWF expression specifically. 

left vein thrombosis:  should specify leg, if that's where it was

Anamnestic means a delayed and enhanced immune response to an antigen- not sure how it applies to family history, as in Figure 1

Figure 4 typo:  Pre

Barred boxes refers to deceased family members (not died)

Gray for not tested

p 7, line 257 :  slight (not slightly) reduction

p 12 line 370 :  associated with

p 12 line 378 consisting of

Round 2

Reviewer 1 Report

Unfortunately, authors' responses did not address this Reviewer's concerns. The role of miR-143/145 expression in lymphocytes to the levels of FVIII and/or VWF in endothelial cells or in the liver has not been proven. Such deficiency may be demonstrated by measuring FVIII and/or VWF in endothelial cell line after transfecting mir-143/145 or their precursors. Alternatively, the 1000 Genomes/GEUVADIS may be merged with other database that contains levels of FVIII and/or VWF in endothelial cells or in the liver. 

This Reviewer advises authors to seek professional to improve English expression. 

Round 3

Reviewer 1 Report

The manuscript has been greatly improved by incorporating additional in vitro findings from the literature. To address this Reviewer's concerns, authors are directed to the article: Patterns of expression of factor VIII and von Willebrand factor by endothelial cell subsets in vivo, by Pan et al., Blood 128(1):104-109, 2016, which contains relevant findings and/or databases to add information and to conduct additional analyses that would strengthen the current manuscript. 

Additional revision is needed to address this Reviewer's concerns.

Round 4

Reviewer 1 Report

Authors should incorporate their comments on Pan et al. reference into the manuscript because the reference provides expression levels of factor VIII and von Willebrand factor across various tissues, including lymphoid tissue, which is most relevant to author's study using lymphoblast cells.

Please check gramma and expression after revision.
